# Mismatch repair deficiency predicts response to HER2 blockade in HER2-negative breast cancer

Nindo B. Punturi[1], Sinem Seker[1], Vaishnavi Devarakonda[2,3], Aloran Mazumder[1], Rashi Kalra[2,3], Ching Hui Chen [2,3], Shunqiang Li[4], Tina Primeau[4], Matthew J. Ellis[2,3], Shyam M. Kavuri [2,3 ✉] & Svasti Haricharan [1 ✉]

Resistance to endocrine treatment occurs in ~30% of ER$^+$ breast cancer patients resulting in ~40,000 deaths/year in the USA. Preclinical studies strongly implicate activation of growth factor receptor, HER2 in endocrine treatment resistance. However, clinical trials of pan-HER inhibitors in ER$^+$/HER2$^-$ patients have disappointed, likely due to a lack of predictive biomarkers. Here we demonstrate that loss of mismatch repair activates HER2 after endocrine treatment in ER$^+$/HER2$^-$ breast cancer cells by protecting HER2 from protein trafficking. Additionally, HER2 activation is indispensable for endocrine treatment resistance in MutL$^-$ cells. Consequently, inhibiting HER2 restores sensitivity to endocrine treatment. Patient data from multiple clinical datasets supports an association between MutL loss, HER2 upregulation, and sensitivity to HER inhibitors in ER$^+$/HER2$^-$ patients. These results provide strong rationale for MutL loss as a first-in-class predictive marker of sensitivity to combinatorial treatment with endocrine intervention and HER inhibitors in endocrine treatment-resistant ER$^+$/HER2$^-$ breast cancer patients.

[1] Tumor Microenvironment and Cancer Immunology, Sanford Burnham Prebys Medical Discovery Institute, La Jolla, CA, USA. [2] Lester and Sue Smith Breast Center, Baylor College of Medicine, Houston, TX, USA. [3] Department of Medicine, Baylor College of Medicine, Houston, TX, USA. [4] Department of Medicine, Washington University in St. Louis, St. Louis, MO, USA. ✉email: kavuri@bcm.edu; sharicharan@sbpdiscovery.org

Estrogen receptor positive (ER$^+$) breast cancer is one of the most common cancers in women worldwide[1]. ER$^+$ breast cancer patients are treated with endocrine therapy, which interrupts ER signaling[2]. A subset of ER$^+$ breast tumors also amplify the tyrosine kinase receptor and oncogene, HER2[3,4]. These ER$^+$/HER$^+$ breast cancer patients are less responsive to endocrine therapy but respond extremely well to combinatorial treatment with HER inhibitors, a seminal discovery[5]. However, the majority of ER$^+$ breast cancer is HER2$^-$ at diagnosis, and while ~70% of ER$^+$/HER2$^-$ breast cancer patients respond well to endocrine therapy, ~30% of patients become resistant to endocrine treatment resulting in relapse, metastasis, and death[2,6].

The discovery that HER2 amplification induces endocrine therapy resistance in ER$^+$ breast cancer spurred research into other means of HER2 activation. These studies identified mutation and phosphorylation as mechanisms by which ER$^+$ HER2 non-amplified (henceforth referred to as ER$^+$ HER2$^-$) breast cancer cells could activate HER2 signaling to resist endocrine treatment[4,7]. However, translation of these findings proved challenging with results from clinical trials failing to live up to preclinical promise[8,9]. There is recognition now that this is likely because only a subset of ER$^+$ breast cancers activate HER2 to resist endocrine therapy. Finding this subset is complicated by the fact that ER$^+$/HER2$^-$ breast cancer cells likely activate HER2 only in response to endocrine therapy, making identification of these patient cohorts from diagnostic biopsies challenging.

Without identifying this patient subset, it is difficult to design a clinical trial with sufficient resolution to uncover real improvement in patient outcome.

Continuing efforts to identify alternate therapies for endocrine-therapy-resistant ER$^+$/HER2$^-$ breast cancer patients have largely failed to show real improvement in the clinic. The only targeted therapy to prove effective to date is CDK4/6 inhibitors[10]. However, these inhibitors have to be administered constantly to be effective, and are, therefore, a financially and physically costly treatment modality that postpones resistance, metastasis, and death but does not remove this threat[11]. Moreover, some endocrine-therapy-resistant patients do not respond to CDK4/6 inhibitors at all[12]. Hope of curing endocrine-therapy-resistant patients with HER2 inhibitors, therefore, remains a tantalizing challenge with clinical impact.

Defects in the MutL complex of mismatch repair, comprised of *MLH1* and *PMS2*, were recently identified as drivers of endocrine treatment resistance in 15–17% of ER$^+$/HER2$^-$ breast cancer patients[13,14]. Mismatch repair is a fundamental DNA repair pathway conserved between pro- and eukaryotes, and essential for guarding the genome during cellular replication[15]. Here, we demonstrate a non-genomic role for MutL loss in activating HER2 in ER$^+$ HER2$^-$ cells exposed to endocrine therapies. Moreover, using multiple experimental model systems, we provide strong evidence for MutL loss as a stratifier of response to HER inhibitors in endocrine-therapy-resistant, nominally HER2$^-$ ER$^+$ breast cancer patients.

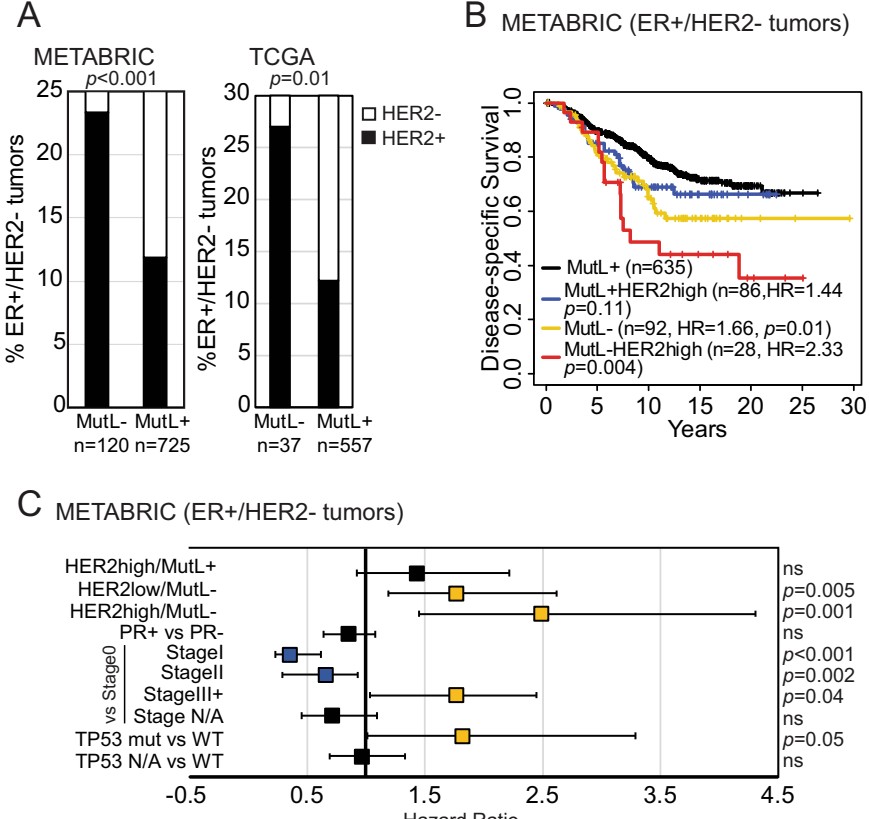

**Fig. 1 ER$^+$, HER2$^-$ (non-amplified) breast cancer patients whose tumors are MutL$^-$ have elevated RNA levels of *HER2* and associate with significantly worse disease-specific survival. A** Incidence of tumors with elevated *HER2* RNA levels within MutL$^-$ and MutL$^+$ ER$^+$/HER2$^-$ breast tumors from METABRIC ($p = 0.0006$) and TCGA. Pearson Chi-Square test identified $p$ values. Corresponding RPPA data in Fig. S2A and contextualization with HER2$^+$ subset in Fig. S2B, C. Kaplan–Meier survival curves (**B**) and proportional hazard assessment (**C**) demonstrating differences in disease-specific survival between specified groups within the ER$^+$/HER2$^-$ breast tumor cohort from METABRIC. Boxes in (**C**) indicate the hazard ratio calculated using the Cox Proportional Hazards Regression analysis and error bars indicate the 95% confidence interval. Stage I $p$ value = 0.0003. Supporting data from TCGA presented in Fig. S2D and proliferation controls in Fig. S2E, F. All statistical tests were two-sided. Source data for this figure are available with paper.

## Results

**Loss of mismatch repair associates with HER2 activation in HER2⁻ breast cancer cells**. To understand mechanisms underlying MutL loss-induced endocrine treatment resistance, we analyzed previously generated reverse phase protein array (RPPA) data to compare ER⁺/HER2⁻ MCF7 breast cancer cells engineered to carry shRNA against *MLH1* or *PMS2* against control isogenic cells with shRNA against Luciferase[13]. This model system has been extensively validated using orthogonal approaches, with pooled RNAi and with rescue using wild-type *MLH1* and is continually revalidated[13,14]. Analysis of the RPPA data identified significant upregulation of phosphorylated HER2 (pHER2) in response to endocrine treatment (fulvestrant) in sh*MLH1* and sh*PMS2* MCF7 cells but not in sh*Luc* cells (Fig. S1). To test whether an association between MutL loss and HER2 activation is also detectable in patient tumors, we analyzed HER2 protein levels from RPPA data in ER⁺ breast tumors that were nominally HER2⁻ (non-amplified) from TCGA. We observed that ~70% of MutL⁻ patient tumors have positive HER2 levels compared to ~50% of MutL⁺ patient tumors (Fig. S2A). These tumor samples are largely treatment-naïve, and therefore correspond more closely to the RPPA data generated from vehicle-treated controls in our model system, where we observe modest upregulation of HER2 protein levels, than to the more robust HER2 upregulation observed in fulvestrant-treated samples (Fig. S1).

Encouraged by this observation, we compared RNA levels using gene expression microarray data from two independent patient tumor datasets: METABRIC and TCGA. We chose to compare RNA levels as these data are more abundant in multiple datasets and permit correlations with patient outcomes. In both cases, we observed that ~25% of MutL⁻ ER⁺/HER2⁻ patient tumors have relatively high RNA levels of HER2 compared to ~10% of MutL⁺ patient tumors (Fig. 1A). While neither RNA nor protein levels in this heterogeneous collection of treatment-naïve and pre-treated patient tumors are as high as that seen in HER2⁺ breast cancer (contextualized in Fig. S2B, C), nonetheless they consistently show modest increase in total HER2 RNA and protein levels in MutL⁻ ER⁺/HER2⁻ patient tumors.

MutL⁻ patient tumors with relatively high *HER2* RNA also associate with significantly worse disease-specific survival in METABRIC (Fig. 1B) and in TCGA (Fig. S2D). Upregulation of *HER2* in MutL⁻ patient tumors also independently prognosticates worse disease-specific survival in Cox Proportional Hazards analyses when considering PR status, tumor stage, and *TP53* mutational status as confounding variables (Fig. 1C). MutL loss as assayed by low gene expression levels is not an artifact of low basal proliferation since RNA levels of *MKI67* (a proliferation marker) are either higher in MutL⁻ patient tumors, or comparable between MutL⁻ and MutL⁺ patient tumors (Fig. S2E, F). Together, these data suggest that the association between MutL loss and HER2 upregulation is of clinical relevance.

**Inhibition of mismatch repair activates HER2 in response to endocrine treatment in ER⁺/HER2⁻ breast cancer cells**. We next tested the causality of this relationship in two independent cell line models of ER⁺/HER2⁻ breast cancer: MCF7 and T47D. Data from these experimental model systems mirror that observed in patient datasets. In both cell lines, Western blotting identified higher baseline levels of pHER2 in cells with stable knockdown of *MLH1* (sh*MLH1*), the principal component of the MutL complex, relative to isogenic MLH1-proficient (sh*Luc*) cells, with further increase upon treatment with ER degrader, fulvestrant (Figs. 2A and S3A). Downstream signaling to pAkt and pS6k is also upregulated in sh*MLH1* cells after fulvestrant

treatment (Fig. 2A). In addition, we confirmed increased HER2 protein at the membrane of sh*MLH1* cells after fulvestrant treatment using both immunofluorescence (Fig. 2B) and flow cytometry (Fig. S3B, C). Increase in membrane HER2 in sh*MLH1* cells after exposure to endocrine treatment was consistent in xenograft tumors from MCF7 sh*Luc* and sh*MLH1* cells (Fig. 2C). This increase in membrane-bound HER2 remained consistent with use of antibodies against either total HER2 (Fig. 2B) or against pHER2 (Figs. 2C and S3E). Also, the same increase in membrane HER2 levels after fulvestrant treatment was seen in tumors from an ER⁺/HER2⁻ patient-derived xenograft (PDX) model of MutL loss (WHIM20[13,16]) (Fig. 2D). We did not observe changes in levels of MLH1 protein in response to HER2 inhibition, validating the directionality of the observed relationship (Fig. S3F). These data indicate that MutL loss directly activates HER2 signaling in ER⁺/HER2⁻ breast cancer cells upon endocrine treatment.

**MutL⁻ cells engage HER2 signaling by protecting HER2 from lysosomal protein trafficking**. Since MutL⁻ ER⁺/HER2⁻ tumors have higher mutation load than MutL⁺ tumors[13,17], we tested whether *HER2* activation in these tumors occurs via activating mutations in *HER2*, a previously established mechanism of HER2 activation in HER2 non-amplified cancer cells[7]. We found no enrichment for *HER2* mutations in ER⁺/HER2⁻ MutL⁻ primary patient tumors relative to MutL⁺ ones (TCGA: 0 vs. 1.8%, METABRIC: 2.2% vs. 2.8% in MutL⁻ vs. MutL⁺). Further, HER2 activation induced by loss of MLH1 in our experimental model systems is reversible when MLH1 is re-expressed in sh*MLH1* cells (Fig. S3D), arguing against an irreversible mutational change as the underlying mechanism. In addition, acute loss of MLH1 by transient transfection of parental MCF7 cells with sgRNA against *MLH1* immediately upregulates pHER2 to similar levels as those seen in cells with stable knockdown of *MLH1* (Fig. S3E). These data both confirm the specificity of the link between MLH1 loss and HER2 activation and argue against an underlying mechanism of mutagenesis. This suggests that MutL loss activates HER2 through non-mutational mechanisms. To identify alternate mechanisms by which MutL loss activates HER2 signaling in conjunction with endocrine treatment, we conducted RNAseq analysis of sh*MLH1* MCF7 cells relative to isogenic sh*Luc* controls at baseline and after fulvestrant treatment (Supplementary Data 1). RNAseq analysis of signatures identified significant enrichment of protein trafficking pathways in MutL⁻ relative to MutL⁺ cells after fulvestrant treatment (Fig. 3A). We found similar enrichment for autophagy and protein trafficking pathways in Reactome analysis of RPPA data comparing MutL⁻ and MutL⁺ cells after fulvestrant treatment (Fig. S4A). Therefore, we next tested whether loss of the MutL complex prevents the targeting of HER2 for lysosomal degradation after endocrine therapy in ER⁺/HER2⁻ breast cancer cells.

First, we conducted a time course immunofluorescence experiment testing colocalization of HER2 with the lysosomal marker, LAMP1 in MCF7 and T47D sh*Luc* and sh*MLH1* cells at baseline and at 18, 36, and 54 h post treatment with fulvestrant. At baseline and at 18 h post treatment, both sh*Luc* and sh*MLH1* cells exhibit low levels of HER2, however by 36 h post treatment, HER2 positivity increases in both cell types. However, 60–80% of sh*Luc* cells with HER2 expression demonstrate colocalization of HER2 with LAMP1 (Figs. 3B and S4B). By 54 h post treatment, HER2 continues to colocalize with LAMP1 in sh*Luc* cells, whereas sh*MLH1* counterparts have HER2 at the membrane, distinct from the perinuclear LAMP1 immunostain (Fig. 3B). Next, we used chloroquine, a known autophagy inhibitor[18,19], to test whether inhibition of lysosomal degradation pathways in

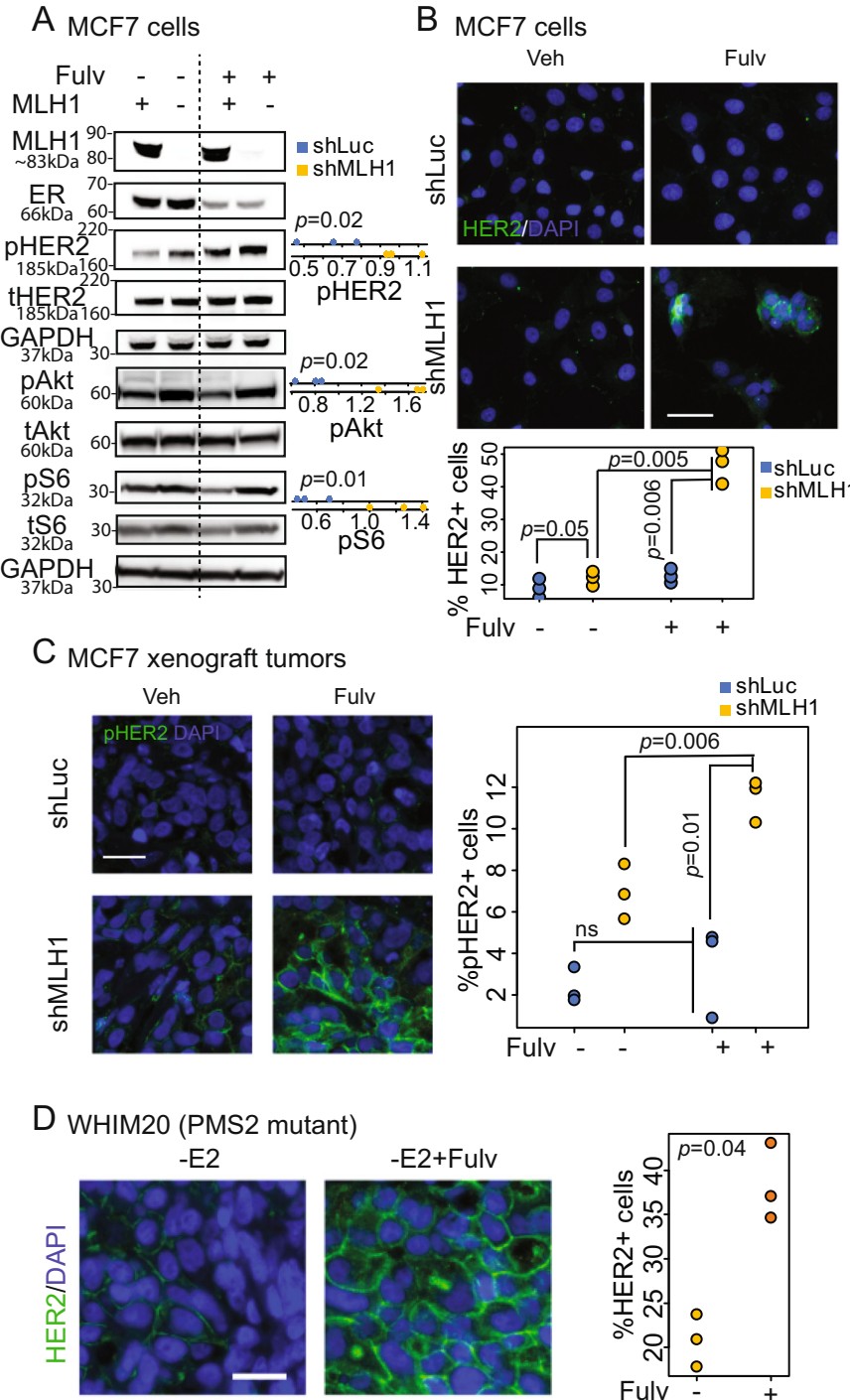

**Fig. 2 MLH1 loss in ER$^+$, nominally HER2$^-$ breast cancer cells upregulates membrane-bound HER2. A** Western blots demonstrating increase in pHER2 and downstream signaling in sh*MLH1* MCF7 cells treated with fulvestrant relative to sh*Luc* cells. Quantification of four independent replicates conducted through ImageJ in accompanying dot plots. Validation in T47D cells in Fig. S3A. Immunofluorescent staining for HER2 in MCF7 sh*Luc* and sh*MLH1* cells in vitro (**B**), in MCF7 sh*Luc* and sh*MLH1* xenograft tumors (**C**), and in WHIM20, *PMS2* mutant, ER$^+$/HER2$^-$ PDX tumors (**D**), grown with or without fulvestrant. Accompanying quantification presented as strip charts. Three independent experiments or tumors from each group were quantified. Two-sided Student's *t* test determined *p* values. Supporting data from FACS analysis are presented in Fig. S3B, C. Scale bars represent 50 μ. Source data for all figures available with paper.

sh*Luc* cells can rescue HER2 positivity after endocrine therapy. In both MCF7 and T47D cells, sh*Luc* cells treated with a combination of fulvestrant and chloroquine demonstrate significant increase in membrane HER2 positivity relative to those treated with fulvestrant alone (Figs. 3C and S4C). Indeed, membrane HER2 positivity is at levels comparable to that of

sh*MLH1* counterparts in both cell lines tested, with the addition of chloroquine.

Finally, we directly tested whether MutL loss prevents targeting of HER2 to autophagosomes by assessing colocalization of transiently transfected *HER2-GFP* and *LC3-RFP* for up to 36 h after administration of fulvestrant in sh*Luc* and sh*MLH1* MCF7

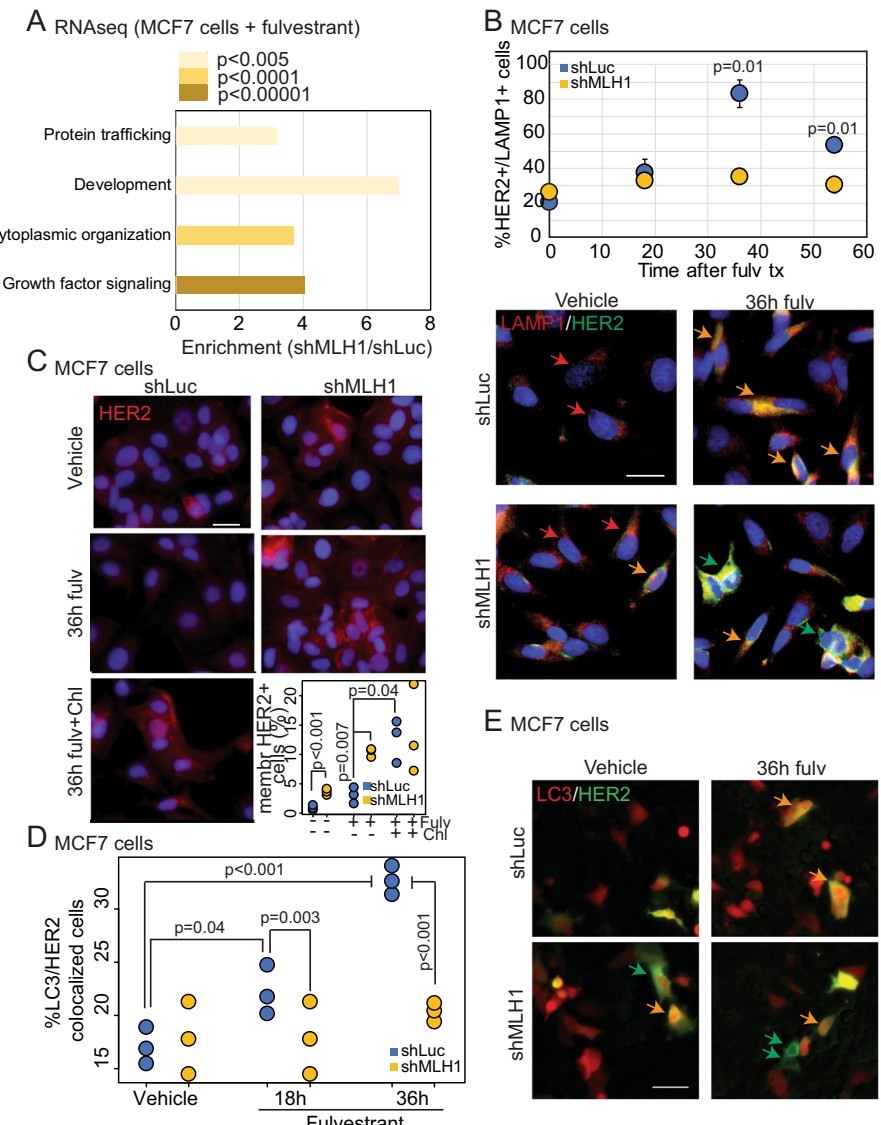

**Fig. 3 MLH1 regulates protein trafficking of HER2. A** Gene set enrichment analysis of RNAseq data comparing isogenic sh*Luc* and sh*MLH1* MCF7 cells after treatment with fulvestrant for 4 days. *P* values were generated using DESeq2 R package and adjusted for multiple comparison using Benjamini–Hochberg. Comparable RPPA data analysis in Fig. S4A. Raw read counts available as supplementary data. **B** Co-immunofluorescence for HER2 and lysosomal marker, LAMP1 (orange arrows) at baseline and after 18, 36, and 54 h of fulvestrant treatment. Green arrows indicate HER2 that is not colocalized with LAMP1, and red arrows indicate LAMP1 positivity alone. Validation in T47D cells in Fig. S4B. **C** Immunofluorescence staining for HER2 in MCF7 sh*Luc* and sh*MLH1* cells treated with vehicle and 36 h of fulvestrant alone or a combination of fulvestrant and chloroquine, an autophagy inhibitor. For sh*Luc* vs. sh*MLH1* vehicle, *p* = 0.0006. Validation in T47D cells in Fig. S4C. Quantification (**D**) and representative photomicrographs (**E**) from 36 h of live cell tracking of colocalization of HER2 and LC3 (orange arrows), a marker of autophagosomes, in MCF7 sh*Luc* and sh*MLH1* cells treated with vehicle or fulvestrant. Green arrows indicate HER2$^+$ LC3$^-$ cells. Cells were tracked after administration of fulvestrant. All quantification is of three independent biological replicates conducted through ImageJ and is represented as strip charts. Two-sided Student's *t* test determined all *p* values. For sh*Luc* vehicle vs. 36-h fulvestrant treatment, *p* = 0.0003 and for sh*Luc* vs. sh*MLH1* at 36-h fulvestrant treatment, *p* = 0.0005. Scale bars represent 50 μ. Source data for all figures available with paper.

cells. While sh*Luc* cells demonstrate increasing colocalization of HER2 and LC3 with time after treatment with fulvestrant, sh*MLH1* cells do not (Fig. 3D, E). In fact, by 36 h after fulvestrant treatment, sh*MLH1* cells with defined membrane HER2 staining are detectable with no LC3 colocalization, whereas this is undetectable in sh*Luc* counterparts (green arrows, Fig. 3E). Together, these data indicate that both sh*Luc* and sh*MLH1* ER$^+$/HER2$^-$ breast cancer cells upregulate HER2 upon ER degradation through endocrine therapy. However, while sh*Luc* cells rapidly target HER2 to lysosomal protein trafficking, sh*MLH1* cells maintain HER2 at the membrane, thereby upregulating

HER2-mediated signaling and inducing endocrine therapy resistance.

**HER2 is required for endocrine treatment resistance of MutL$^-$ ER$^+$/HER2$^-$ breast cancer cells.** To test whether HER2 activation in MutL$^-$ cells is required for endocrine-therapy-resistant growth, we used siRNA to decrease endogenous *HER2* in MCF7 sh*Luc* and sh*MLH1* cells, and then assayed growth in presence of fulvestrant. We observed complete rescue of endocrine treatment sensitivity in sh*MLH1* cells transfected with si*HER2*, with no observable change in endocrine therapy response

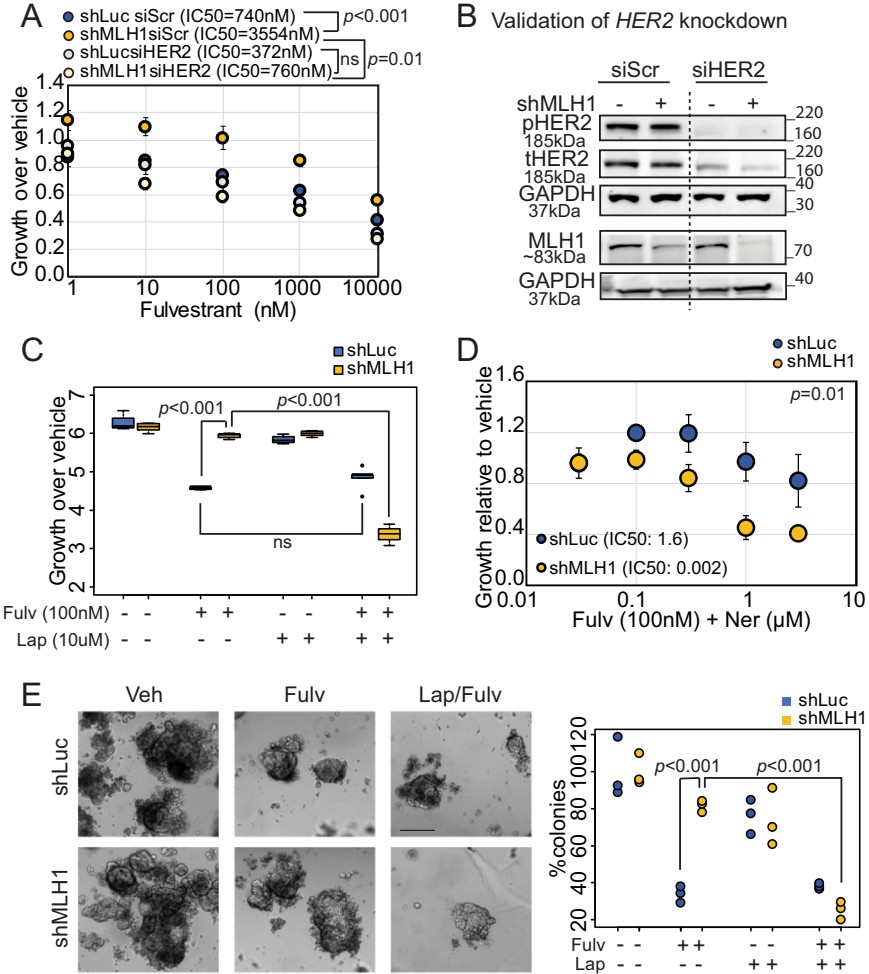

**Fig. 4 HER2 is required for endocrine-therapy-resistant growth of ER$^+$ MLH1$^-$ breast cancer cells.** Knockdown of endogenous *HER2* using siRNA against *HER2* or a scrambled control in MCF7 sh*Luc* and sh*MLH1* cells validated by Western blotting (**B**) and followed by 2D growth assays for dose response to fulvestrant treatment (**A**). For sh*Luc* vs. sh*MLH1* with siScr, $p = 0.0001$. Supporting data demonstrating similar response to tamoxifen and estrogen deprivation in Fig. S5A, B. **C** Growth of MCF7 sh*Luc* and sh*MLH1* cells in response to specified therapeutic combinations represented as a bar graph. For sh*Luc* vs. sh*MLH1* with fulvestrant treatment, $p = 0.0009$, and for sh*MLH1* fulvestrant vs. lapatinib + fulvestrant treatment, $p = 3.15e - 05$. Supporting data from T47D in Fig. S5C. **D** Dose curve demonstrating response to neratinib and fulvestrant in MCF7 sh*Luc* and sh*MLH1* cells. Supporting data demonstrating similar results in T47D cells and in response to tamoxifen in Fig. S5D, E. **E** 3D growth in Matrigel of MCF7 sh*Luc* and sh*MLH1* cells in response to specified treatments. Representative images for each treatment group (except lapatinib, which showed no visible difference from vehicle-treated) shown alongside quantification. For both sh*Luc* vs. sh*MLH1* fulvestrant-treated, and sh*MLH1* fulvestrant vs. fulvestrant + lapatinib-treated, $p = 0.0002$. Supporting data in T47D cells in Fig. S6A. For dose curve experiments **A, C, D** three independent biological replicates were quantified for each group and each dose point. IC50 values were determined over three independent experiments and compared for statistical differences. Circles (**A**, **D**) represent mean growth relative to vehicle-treated cells over 7 days of treatment and error bars the standard deviation. Box plots show median, quartiles, minima and maxima, and outliers at 1.5× IQR. All statistical comparisons used the two-sided Student's *t* test. All experiments were conducted >2 times. Source data for this figure available with paper.

in sh*Luc* cells under the same conditions (Fig. 4A, B). This rescue of sensitivity to endocrine therapy extends to tamoxifen (Fig. S5A) and estrogen deprivation, a surrogate for aromatase inhibitors (Fig. S5B). In keeping with this observation, both MCF7 (Fig. 4C) and T47D (Fig. S5C) sh*MLH1* cells grown in 2D are sensitive to combinatorial administration of fulvestrant and lapatinib, a HER inhibitor used in clinic. In addition, MCF7 (Fig. 4D) and T47D (Fig. S5D) sh*MLH1* cells demonstrate increased sensitivity to fulvestrant when treated with neratinib, another HER inhibitor. Similar results were obtained when neratinib was combined with tamoxifen treatment (Fig. S5E).

To test specificity of MLH1 loss in inducing therapeutic vulnerability to HER2 inhibitors, we also tested growth response to lapatinib in two previously established endocrine therapy resistance models: MCF7 cells harboring either an ESR1-YAP1 or

an ESR1-PCDH11X fusion[20]. Both these model systems with no known defects in mismatch repair are resistant to endocrine therapy, fulvestrant, as expected, but remain resistant to lapatinib compared to MCF7 sh*MLH1* cells (Fig. S5F). Finally, both MCF7 (Fig. 4E) and T47D (Fig. S6A) sh*MLH1* cells demonstrated persistent 3D growth relative to sh*Luc* cells in response to fulvestrant, but this growth was significantly suppressed by adding lapatinib.

These data suggest that loss of MutL predisposes ER$^+$/HER2$^-$ breast cancer cells to respond to HER inhibitors in concert with endocrine therapies. To test this proposition in vivo, we randomized mice with MCF7 sh*Luc* and sh*MLH1* xenograft tumors into five treatment arms: control (with estrogen supplementation), estrogen deprivation, fulvestrant (and estrogen deprivation), lapatinib (and estrogen deprivation), and a

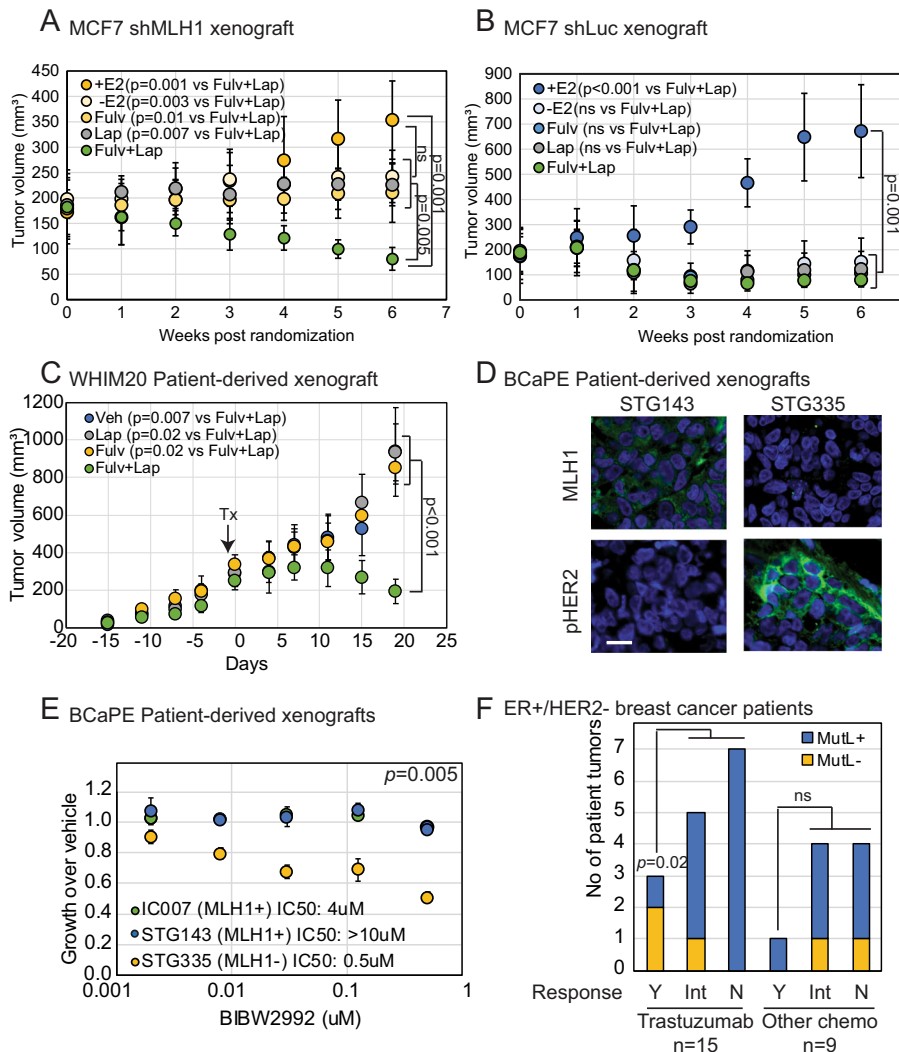

**Fig. 5 MLH1 loss predicts sensitivity to HER inhibitors in endocrine-therapy-resistant ER$^+$, HER2$^-$ breast cancer cells in vivo and in patient tumors.** In vivo xenograft experiments of MCF7 sh*MLH1* (**A**), sh*Luc* (**B**) cells, and WHIM20 PDX line (**C**) demonstrating response in tumor growth to specified treatments. Control group: $n = 5$ mice; estrogen-deprived group: $n = 4$ (**A**), $n = 5$ (**B**), and $n = 7$ (**C**); lapatinib group: $n = 6$ (**A**), $n = 5$ (**B**), $n = 4$ (**C**); fulvestrant group: $n = 5$ (**A**), $n = 6$ (**B**), $n = 3$ (**C**); fulvestrant + lapatinib group: $n = 8$ (**A**), $n = 6$ (**B**), $n = 5$ (**C**). Circles represent the mean and error bars the standard deviation. Student's $t$ test determined $p$ values by comparing slopes. For (**B**), +E$_2$ vs. fulvestrant + lapatinib-treated tumors, $p = 0.0005$ and for (**C**), fulvestrant + lapatinib-treated tumors vs. rest, $p = 4.7e - 05$. **D** Immunofluorescence depicting protein levels of MLH1 in two ER$^+$/HER2$^-$ PDX lines associating with HER2 activation after estrogen deprivation. Scale bar $= 50 \mu$. **E** Growth curves demonstrating sensitivity of these PDX lines to HER inhibition ex vivo. IC50 calculated using regression analysis over three independent experiments and compared using two-sided Student's $t$ test. Circles represent the mean and error bars the standard deviation. Supporting data in Fig. S7A, B. **F** Categorical analysis supporting increased sensitivity of ER$^+$/MutL$^-$ patient tumors to trastuzumab in combination with other chemotherapy. Supporting data in Fig. S7E, F. Y yes, Int intermediate, N no. Two-sided Fisher's Exact test determined $p$ values. For all regression analyses, individual $p$ values and multiple adjusted $R^2$ values were derived using a linear model in R. Source data available with paper.

combination of fulvestrant and lapatinib (and estrogen deprivation). As expected from previous experiments[13], we observed estrogen independent and fulvestrant-resistant growth in MCF7 sh*MLH1* tumors, and little response to lapatinib alone (Fig. 5A). However, there was striking response with tumor shrinkage to the combination of fulvestrant and lapatinib (Fig. 5A). In contrast, MCF7 sh*Luc* tumors demonstrated tumor shrinkage in response to either estrogen deprivation or fulvestrant treatment alone and no further response to the addition of lapatinib (Fig. 5B), in keeping with previous literature[21–23].

**Loss of mismatch repair increases sensitivity to HER inhibitors in vivo and in patient tumors.** We next tested whether MutL defects had similar associations with sensitivity to HER

inhibitors in PDX tumors. In vivo growth of WHIM20, *PMS2* mutant, ER$^+$/HER2$^-$ PDX tumors xenografted into mouse mammary fat pads demonstrated a similar pattern of tumor regression in response to combination of lapatinib and fulvestrant but not in response to either treatment alone (Fig. 5C). To test whether loss of PMS2 causally activates HER2 and induces response to HER2 inhibitors, similarly to MLH1, we tested our previously established and validated MCF7 cells with stable knockdown of *PMS2*[13] (Fig. S6B). We observed high baseline levels of pHER2 in sh*PMS2* cells and further induction after fulvestrant treatment (Fig. S6C). This upregulation of HER2 levels was reflected in increased sensitivity to HER inhibitor, lapatinib, in sh*PMS2* MCF7 cells relative to isogenic sh*Luc* cells (Fig. S6D).

An additional ER$^+$/HER2$^-$ PDX line[24] with low MLH1 protein (Fig. 5D) and low *MLH1* RNA levels (Fig. S7A) also has increased membrane-bound HER2 (Fig. 5D). This increase in HER2 protein at the membrane also associates with increased sensitivity to BIBW2992 (or afatinib, a second generation pan-HER inhibitor), as assayed by ex vivo 3D growth (Fig. 5E). We also observed significant correlation between sensitivity to three HER inhibitors, including lapatinib, and low RNA levels of *MLH1*/*PMS2* across seven PDX models of luminal breast cancer[24] grown in estrogen-deprived conditions (Fig. S7A). Of note, there was no such correlation across 11 PDX models of basal-like breast cancer (Fig. S7B). Together, these data demonstrate that MutL loss predisposes ER$^+$/HER2$^-$ PDX tumors to respond to a combination of HER inhibitors and endocrine treatment.

We also validated our findings in transcriptomics data from ER$^+$/HER2$^-$ patient tumors biopsied at diagnosis and after 4–6 weeks of neoadjuvant aromatase inhibitor treatment (Z1031[25]). We first confirmed inverse association between RNA levels of *HER2* and *MLH1* in these tumors at diagnosis (Fig. S7C), indicating that tumors with low *MLH1* have relatively higher *HER2* at baseline (as observed in our experimental model systems and in TCGA and METABRIC patient tumor datasets). Next, we identified direct association between RNA levels of *HER2* and proliferation as measured by immunohistochemistry for Ki67 after endocrine treatment (Fig. S7C). Importantly, this association is restricted to MutL$^-$ tumors and not seen in tumors that are MutL$^+$ (Fig. S7C).

These data suggest that loss of MutL induces HER2-associated proliferation in ER$^+$/HER2$^-$ breast cancer cells treated with endocrine intervention. As an additional control, we found no significant associations between levels of *HER2* RNA and those of another mismatch repair gene, *MSH2*, which is not part of the MutL complex (Fig. S7D). This specificity increases confidence in the association between HER2 activation and MutL loss. Second, association between *HER2* RNA levels and Ki67 in MutL$^-$ ER$^+$/HER2$^-$ breast tumors is only significant after exposure to endocrine treatment and not in pre-treatment biopsies (Fig. S7D). We confirmed that loss of MutL in patient tumors is not merely a consequence of low proliferation (Fig. S7E). This attests to the role of endocrine therapy in catalyzing reliance on HER2 for proliferation in MutL$^-$ ER$^+$/HER2$^-$ tumors.

We also analyzed a second dataset[26] where metastatic, treatment-resistant breast cancer patients, irrespective of HER2 status, were randomized to two arms of treatment: anthracyclines and taxanes or anthracyclines, taxanes and trastuzumab, a HER2 inhibitor. From this dataset, we parsed the subset of patients whose cancer was ER$^+$/HER2$^-$ for further analysis. Strikingly, all patients with MutL$^-$ ER$^+$/HER2$^-$ breast cancer demonstrate at least partial response to trastuzumab, compared to less than half of patients with MutL$^+$ ER$^+$/HER2$^-$ disease (Fig. 5F). In addition, 2/3rd of MutL$^-$ patients has complete response to the trastuzumab combination compared to less than a tenth of MutL$^+$ patients (Fig. 5F). This disparate response was only observed in the treatment group where trastuzumab was added to the chemotherapy administered to patients. Concomitant downregulation of *HER2* RNA in response to the trastuzumab combination, but not in response to anthracyclines/taxanes alone was confirmed in the MutL$^-$ ER$^+$/HER2$^-$ tumors (Fig. S7F). These data, while of small sample size, provide support for a role for MutL loss in sensitizing endocrine-therapy-resistant ER$^+$/HER2$^-$ breast cancer to a combination of HER inhibitors and endocrine therapy.

## Discussion

Taken together, results presented here suggest that *MLH1*/*PMS2* downregulation could constitute a first-in-class predictive marker

for response to HER2 inhibition in endocrine-therapy-resistant ER$^+$/HER2$^-$ breast cancer. The only other biomarkers proposed to predict response to HER2 inhibitors in the endocrine-therapy-resistant ER$^+$/HER2$^-$ setting are low ER/PR but these markers are not specific to HER2 activation and have mixed associations across clinical trials decreasing their feasibility for clinical use[8,27]. The impact of the discovery presented here could be substantial, given that loss of nuclear MLH1 and PMS2 occurs in 15–17% of ER$^+$/HER2$^-$ breast cancer[28]. Importantly, it is clinically feasible to assess these markers at diagnosis to tailor therapy since diagnostic assays for MLH1 and PMS2 loss are routinely implemented in clinic for colorectal and endometrial cancer patients[29,30]. Coopting these diagnostic tests for ER$^+$/HER2$^-$ breast cancer patients is, therefore, relatively straightforward and could benefit a significant subset of patients.

The mechanism underlying HER2 activation in response to endocrine therapy in MutL$^-$ ER$^+$/HER2$^-$ breast cancer cells is through lysosomal protein trafficking. Data from Western blots and immunofluorescence of cell lines and PDX tumors suggest that total HER2 levels increase with MutL loss even before endocrine therapy. Concordantly, baseline levels of HER2 appear higher in ER$^+$/HER2$^-$ MutL$^-$ breast cancer cells in patient tumor gene expression data, although many orders lower than levels in HER2-amplified patient tumors. These data suggest that even at baseline, MutL$^-$ cells are less likely to target HER2 for protein degradation. However, with endocrine therapy, HER2 is upregulated in both MutL$^+$ and MutL$^-$ ER$^+$/HER2$^-$ breast cancer cells as predicted by the literature[3]. In the context of this HER2 upregulation, the protection of HER2 from protein trafficking in MutL$^-$ cells allows HER2-mediated growth signaling to compensate as a cell-cycle cue for the loss of ER driven by standard endocrine therapies. These data provide an explanation for the lack of positive clinical trial data from using HER inhibitors in the endocrine-therapy-resistant ER$^+$/HER2$^-$ breast cancer setting, in spite of strong preclinical links between ER loss and upregulation of HER2[3,4]. The link between loss of a DNA damage repair pathway and targeting of growth factor proteins for protein trafficking requires further investigation.

A significant limitation of this study is the lack of specific clinical trial data with which to test the hypothesis raised by the molecular biology described above. Very few clinical trials have been performed to test efficacy of HER inhibitors in ER$^+$/HER2$^-$ breast cancer patients[8,31]. None of these trials include transcriptomic or genomic data accrual from tumor biopsies and since MutL status is not routinely tested in clinic for breast cancer patients, this data is missing from all existing trials. The strength of preclinical data presented here and the strong associations observed in the limited clinical trial data available make a compelling case for revisiting HER inhibitors in the ER$^+$/HER2$^-$ breast cancer setting but this time in context of MutL status.

These results also have significance beyond ER$^+$ breast cancer. Our data provide support for a recent report on Lynch syndrome colorectal cancer suggesting a link between loss of mismatch repair and response to HER inhibitors[32]. Lynch syndrome is one of the most common causes of inherited cancers at many sites and is caused by hereditary defects in mismatch repair genes[33]. In addition, mismatch repair loss drives a significant proportion of sporadic colorectal, ovarian, and endometrial cancer[34]. If *MLH1*/*PMS2* loss serves as a predictive marker for sensitivity to HER inhibitors across cancer types, the already routine identification of these markers in these other cancer types can be married to a clinically feasible targeted therapy.

## Methods

**Cell lines, mice, CRISPR, si/shRNA transfection, and growth assays**. Cell lines were obtained from the ATCC (2015) and maintained and validated as previously

reported[35]. *Mycoplasma* tests were performed on parent cell lines and stable cell lines every 6 months (latest test: 02/19) with the Lonza Mycoalert Plus Kit (cat# LT07-710) as per the manufacturer's instructions. Cell lines were discarded at <25 passages, and fresh vials were thawed out. Key experiments were repeated with each fresh thaw. Transient transfection with siRNA against HER2 was conducted using JetPrime PolyPlus transfection reagent[35], and siRNA pools were purchased from Sigma-Aldrich. Stable cell lines were maintained in presence of specified antibiotics at recommended concentrations. Knockdown was validated using qRT-PCR (list of primers used in Supplementary Table 1) and/or Western blotting. Growth assays were conducted in triplicate and repeated independently using Alamar blue to identify cell viability[13]. Growth assay results were plotted as fold change in growth from day 1 to day 7 and normalized as specified. Three-dimensional growth assays were conducted over 4–6 weeks with weekly drug treatments using standard protocols[7]. Images were captured when colonies had established (at 2 weeks), and then treatment was administered, with images taken again at 1 and 3 weeks post treatment. Fold change in area of colonies was calculated over time and represented as %growth. Tumor growth assays in vivo were carried out by injecting $2–5 \times 10^6$ MCF7 cells into the L4 mammary fat pad/mouse. Mice for the MCF7 experiments were 4- to 6-week athymic nu/nu female mice (Envigo or SBP animal facility). For WHIM20 PDX experiments, 6- to 8-week female SCID/Bg mice were purchased from Jackson laboratory. Tumor volume was measured twice or thrice weekly using calipers to make 2 diametric measurements. Tumors were randomized for treatment at 50–150-mm$^3$ volume for MCF7 xenografts and 100–300-mm$^3$ volume for WHIM20 PDX experiments. Tumors were harvested at <2-cm diameter and were embedded in paraffin blocks, OCT, and snap-frozen[36]. Mice that died within 3 weeks of tumor growth rate experiments were excluded from analysis. For all mouse experiments, investigator was blinded to groups and to outcomes. STG335, STG143, and VHIO244 PDX experiment results were kindly provided by the BCaPE consortium, but tumor sections were stained in house. All mouse experiments were performed in compliance with all relevant ethical regulations for animal testing and research, and all experiments conducted in the study received approval from the respective Institutional Animal Care and Use Committee boards (protocols# AN-6934 for Baylor College of Medicine and 18-065 for Sanford Burnham Prebys).

**Inhibitors and agonists.** All drugs were maintained as stock solutions in DMSO, and stock solutions were stored at −80 and working stocks at −20 unless otherwise mentioned. 4-OHT (Sigma-Aldrich, cat# H7904) and fulvestrant (SelleckChem, cat# I4409) were purchased, and stocks were diluted to 10-mmol/L working stocks for all experiments other than dose curves, where specified concentrations were used. For all experiments, cells were treated 24 h after plating, and thereafter every 48 h until completion of experiment. For mouse xenograft experiments, fulvestrant concentrations of 250-mg/kg body weight were prepared in corn oil, freshly on day of injection and administered subcutaneously. Beta-estradiol was purchased from Sigma-Aldrich (cat# E8875), maintained in sterile, nuclease-free water, and diluted to obtain 10-mmol/L stocks for in vitro experiments. For mouse xenograft experiments, 17 β-estradiol was maintained in 200-proof ethanol at 2.7-mg/ml stock solution and added to drinking water twice a week at a final concentration of 8 μg/mL (cat# E2758; Sigma). For experiments involving Chloroquine (Selleckchem, cat#S4157), cells were treated at 50 μM for 16 h before end of assay. Lapatinib (SelleckChem, cat#S2111) and Neratinib were used at specified concentrations. Lapatinib tablets were used at 100 mg/kg in chow from Research Diets, Inc for tumor growth assays.

**Flow cytometry, immunostaining, and microscopy.** Flow cytometry for membrane-bound HER2 was performed based on manufacturer's instructions. After fulvestrant treatment, cells were detached from plates using StemPro™ Accutase™ Cell Dissociation Reagent (cat#A1110501). Cells were washed with chilled PBS and suspended in antibody solution, as per the manufacturer's instructions, in 5-mL flow cytometry tubes and incubated on ice for 20 min. Live cells were then run through BD Accuri C6 cytometer to assess only membrane-bound HER2 protein levels. IF was performed based on the manufacturer's instructions. Cells were washed in PBS; fixed for 20 min at room temperature in 4% PFA; blocked for 1 h at room temperature in 5% goat serum and 1% Triton X-100 in 1x PBS; incubated with primary antibody overnight at 4° in 1% goat serum and 1% Triton X-100 in 1x PBS antibody diluent; incubated with secondary antibody in diluent for 1 h at RT; and then mounted with DAPI-containing mounting media (cat# P36935). Tumor section staining was done using a standard protocol. Briefly, slides were incubated at 65° for 4 h and deparaffinized. Antigen retrieval was done with 10-mM Sodium Citrate (pH 6) for 25 min in pressure cooker. Hereafter, the cells were treated the same as the 2D IF. Primary antibodies used include pHER2 (EMD millipore; cat# 06-229; 1:200) and Ki67 (Novus Biologics, cat# NB500-170SS, 1:250). Cells were treated with fulvestrant for 24 h before evaluation. Fluorescent images were captured with a Nikon microscope and quantified with ImageJ. Representative images were translated into figures using Adobe Photoshop and Adobe Illustrator.

**RNAseq analyses.** RNAseq data were generated from two replicates each of MCF7 shLuc and shMLH1 cells treated with either vehicle or 100-nM fulvestrant for 4 days on the Illumina NovaSeq platforms with paired-end 150-bp sequencing.

Downstream analysis was performed using a combination of programs including STAR, HTseq, Cufflink, and Novogene's wrapped scripts. Alignments were parsed using STAR program and differential expressions were determined through DESeq2/edgeR. FPKM of each gene was calculated based on the length of the gene and read counts mapped to this gene. GO and KEGG enrichment were implemented by the ClusterProfiler. Source data available in supplementary files.

**Lysosomal analyses.** Immunofluorescence of LAMP1/HER2 was conducted by plating 20k cells per well/per condition in a 96-well plate and treated with 100-nM fulvestrant (SelleckChem, cat# I4409) for 36 h. Cells were then probed ON at 4 °C with LAMP1 (proteintech, cat# 21997-1-AP) and HER2 (Invitrogen, cat# MA5-13105) antibodies used at a 1:750 and 1:250 dilution, respectively, diluted in 1x TBST with 5% Goat Serum. For HER2 and LC3 immunofluorescent images, cells were transiently transfected with mCherry-hLC3B-pcDNA3.1, a gift from David Rubinsztein (Addgene plasmid # 40827; http://n2t.net/addgene:40827; RRID: Addgene_40827) and pCMV3-C-GFPSpark-HindIII-XbaI (SinoBiological, cat# HG10004-ACG) using jetPRIME transfection reagent (Polyplus, cat#114-07) as per manufacturers' instructions. Thirty-six hours 100-nM Fulvestrant treatment started 16 h after transfection. Both assays were imaged using BioTek Cytation 5 Imaging Reader.

**Protein analyses.** Western blotting was conducted as described[35]. Cells were exposed to 18–24 h of fulvestrant treatment administered 40 h after plating. For pHER2 Western blots, primary antibody was incubated for 48 h at 4°. For all other antibodies, primary incubation was 2 h at room temperature. All antibodies diluted in 1x TBST and incubated overnight at 4 °C. Antibodies used were pHER2 Y1196 (D66B7) (Cell Signaling; cat# 6942S), total HER2 (Thermo Scientific; NeoMarkers; cat# MS-730-P1ABX), pAkt S473 (D9E) XP (Cell Signaling; cat#4060S), total Akt (Cell Signaling; cat#9272S), pS6 (S235/236) (Cell Signaling; cat# 2211S), total S6 (5G10) (Cell Signaling; cat# 2217S), MLH1 (1:2,000, Sigma-Aldrich; cat# WH0004292M2), ER clone 60C (EMD Millipore; cat# 04-820), and GAPDH (0411) (Santa Cruz; cat# sc-47724). Unless otherwise specified, primary antibodies were diluted 1:1000 for Western blotting. RPPA assays were carried out as described previously with minor modifications[37].

**Statistical analysis.** ANOVA or Student *t* test was used for independent samples with normal distribution. Where distribution was not normal (assessed using Q–Q plots with the Wilk–Shapiro test of normality), either the Kruskal–Wallis or Wilcoxon Rank Sum test was used. All experiments were conducted in triplicate, and each experiment was duplicated independently >2 times. These criteria were formulated to ensure that results from each dataset were calculable within the range of sensitivity of the statistical test used. Databases used for human data mining are from publically available resources: Oncomine, cBio[38], and COSMIC. Z1031 dataset was used with permission from the Alliance consortium. All patients provided informed consent, and studies were conducted according to ethical guidelines and with Institutional Review Board approval from each of the institutions involved in this previously published study. MutL$^-$ tumor from METABRIC, TCGA, and Z1031 datasets was determined in a case list containing all ER$^+$ sample IDs based on gene expression less than mean—$1.5 \times$ standard deviation and/or the presence of nonsilent mutations in *MLH1* and *PMS2*. For the multivariate analysis, we analyzed ER$^+$ tumor samples, extracting mutation data from the cBio portal, and corresponding clinical data through Oncomine. Only samples with survival metadata were included in the analysis. Gene expression, and survival data for TCGA samples were downloaded from cBio portal. All survival data were analyzed using Kaplan–Meier curves and log-rank tests. Proportional hazards were determined using Cox regression. Sample size for animal experiments was estimated using power calculations in R. *P* values were adjusted for multiple comparisons where appropriate using Benjamini–Hochberg. All graphs and statistical analyses were generated either in MS Excel or R and edited in Adobe Photoshop or Illustrator. Z1031ClinicalTrials.gov Identifier: NCT00265759. Data for Z1031 samples available in dbGaP (phs000472.v2.p1).

**Reporting summary.** Further information on research design is available in the Nature Research Reporting Summary linked to this article.

## Data availability

The patient datasets analyzed during the current study are all publicly available from cBio data portal at cbioportal.org (TCGA and METABRIC), or from Gene Expression Omnibus (GEO) (https://www.ebi.ac.uk/arrayexpress/experiments/E-GEOD-28826/). Z1031 ClinicalTrials.gov Identifier: NCT00265759. Data for Z1031 samples available in dbGaP (phs000472.v2.p1). Raw read count data from RNAseq that support the findings of this study are available in Supplementary data. Source data are provided with this paper.

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

## Acknowledgements

We would like to acknowledge the Patient-derived Xenograft and Advanced In Vivo Models core (funded by P30 Cancer Center Support Grant NCI-CA125123, CPRIT Core Facilities Support Grant RP170691) and Dr Michael T. Lewis, Ph.D., Academic Director, Lacey E. Dobrolecki, MS, Core Director at Baylor College of Medicine for helping us in engrafting WHIM20 PDX explants. We also thank Dr Alejandra Bruna (CRUK, UK) and Dr Violeta Serra (VHIO, Barcelona) for providing PDX drug response data and tumor sections for the STG and VHIO PDX lines. Work in this study was funded by Department of Defense Breast Cancer Research Program Breakthrough awards (W81XWH-18-1-0034 to S.H., W81XWH-18-1-0035 to S.M.K.), NCI K22 Career Development award (CA229613 to S.H.), Susan G. Komen Promise Grant (PG12220321 to M.J.E.), SPORE grant (P50CA186784-06), and Cancer Prevention and Research Institute of Texas (CPRIT) Recruitment of Established Investigators award (RR140033 to M.J.E.), National Cancer Institute of the National Institutes of Health under Award Numbers U10CA180821 and U10CA180882 (to the Alliance for Clinical Trials in Oncology), U24C196171.

## Author contributions

N.B.P. designed and performed experiments, analyzed data, and helped write the manuscript. S.S. helped design, conduct, and analyze data from Western blots and xenograft experiments. V.D. and A.M. helped design and conduct 3D Matrigel assays and immunofluorescence experiments. S.L., T.P., R.K. and C.-H.C. conducted WHIM20 patient-derived xenograft experiment. M.J.E. and S.M.K. helped design experiments and interpret results, and edit the manuscript. S.H. designed and performed experiments, analyzed and interpreted data, and wrote and edited the manuscript.

## Competing interests

M.J.E. has intellectual property ownership and received royalties for the PAM50-based breast cancer test "Prosigna." In the last 5 years he has received ad hoc consulting fees and meals (<$5000 per year) from Abbvie, Novartis, AstraZenica, Pfizer, Sermonix, and Puma. S.M.K. is a stakeholder in NeoZenome Therapeutics Inc. S.L. has received license fee from Envigo. He received research funding from Pfizer, Takeda Oncology, Zenopharm, NIH, and DOD, outside of this project. The other authors declare no competing interests.
