## [Peer Review File · Nature Communications]

Reviewers' Comments:

Reviewer #1:

Remarks to the Author:

This paper investigates the role of mismatch repair genes in HER2+ and ER+ breast cancer. The authors show convincing correlations between MutL status and HER2 positivity in a cohort of patient and link MutL/HER2 expression with survival outcome. The inverse correlation between MutL and HER2 expression was validated using an independent mRNA analysis and again, this was linked with differences in survival. Cell line models were used to assess the role of MLH1 and under conditions where MLH1 was depleted, HER2 levels were shown to become elevated. Response to Fulvestrant (and subsequently, to other ER pathway inhibitors) was shown to occur in MLH1 depleted cells, in combination with a HER2 small molecule inhibitor. This was validated in the same cell line models, grown in 3D. A series of in vivo experiments are undertaken, including cell lines depleted for MLH1, which again show responses in the presence of both a HER2 targeted agent and an ER targeted drug. A number of PDX models with varying genetic and MLH1 levels were used to explore this treatment context.

This is a very interesting topic and the findings have the potential to be of substantial clinical relevance and as such, as a concept, I find the main conclusions to be important and novel. Most of the work is conducted to a high standard and although I understand that the authors conducted a number of different in vivo models, the critical comparison is whether the presence or absence of MLH1 impacts tumor growth and treatment response. This is inferred by using models with endogenously low or high MLH1 levels, but what is missing is the isogenically matched comparison. What is surprising is that the authors have these models, namely the shMLH1 and matched control cell line models, but this experiment is not conducted.

- Why did the authors not repeat the experiment in Figure 4A (the MCF7 cell in vivo experiment) to compare the shMLH1 and the control line. Surely this is the most critical comparison, otherwise there is no in vivo data to show that MLH1 is important in the treatment response.
- What is concerning as well, is that the data presented from the in vivo experiment is a heavily modified version of the data and instead of simply plotting the tumor growth curves side by side, the authors present the data as 'slope of tumor growth' which is a non-standard way of presenting the data and is not intuitive or informative. The fact that the growth curves are not plotted according to standard approaches, leads me to conclude that the data wasn't very convincing. Given that this is the most important experiment (treatments to assess in vivo tumor growth in matched cells with or without MLH1), this piece of data is essential, but at the moment, only one part of the data is presented and it is shown in a way that is opaque and uninformative, at best.
- Why do the HER2 inhibitors only work in conditions where ER is inhibited. This seems like the key mechanistic finding, but no work (or even discussion) is provided. Why does ER need to be inhibited for HER2 targeted drugs to work under MLH1 depleted conditions?
- Is MLH1 regulated by estrogen stimulation or by the HER2 pathway? I am a little concerned that MLH1 levels might correlate with outcome because it tracks with either ER and/or HER2 activity, rather than the other way around (i.e. that ER/HER2 activity and response to inhibitors is influenced by MLH1 expression). The initial correlative data suggests that this is not the case, but as a control, it would be useful to assess whether MLH1 is regulated by the ER and/or HER2 pathways.

Reviewer #2:

Remarks to the Author:

Punturi et al have submitted an intriguing manuscript following on from previously published work showing the MLH1 loss drives endocrine treatment resistance. Here, they show that MutL loss activates HER2 in ER+/HER- breast cancer upon endocrine treatment and inhibiting HER2 can restore sensitivity in these tumours.

They present solid in vivo data confirming the potential therapeutic application of this data. However there are a number of issues that should be addressed prior to publication.

1. There is little data investigating the mechanism behind HER2 activation upon MutL loss upon

endocrine treatment. This needs further elucidation - how is HER expression regulated? Why is it ER dependent upon MutL loss?

2. There is no attempt to decipher whether this effect is mediated via MLH1 or PMS2, both components of the MutL complex and loss of MLH1 leads to reduced PMS2 levels - which would be restored upon MLH1 re-expression, so PMS2 alone as the mediator cannot be ruled out based on the current data.

3. The majority of the data are reliant on limited cell lines (only 2) and also no control cell lines.

4. The majority of the data is using only one shMLH1 to show the phenotype. It is widely known that gene silencing techniques in all forms have off target effects so it is imperative that additional approaches are used to validate the findings eg siRNA, CRISPR-Cas9, additional shRNA constructs

5. There is some inconsistency between Fig 2A and Fig3B in terms of pHER2 expression upon MLH1 loss. In Fig 2A there is visibly less pHER2 upon shMLH1 whilst no difference is observed in Fig 3B.

6. Fig 3B needs an MLH1 blot to confirm silencing of the protein

This revised submission has benefited from the additional experiments performed in response to reviewers (revised Fig 3, revised Fig S4, revised Fig 5B, revised Fig S6). We thank the reviewers for their insightful critiques and have addressed them all comprehensively as outlined below.

Reviewer #1 (Remarks to the Author):

This paper investigates the role of mismatch repair genes in HER2+ and ER+ breast cancer. The authors show convincing correlations between MutL status and HER2 positivity in a cohort of patient and link MutL/HER2 expression with survival outcome. The inverse correlation between MutL and HER2 expression was validated using an independent mRNA analysis and again, this was linked with differences in survival. Cell line models were used to assess the role of MLH1 and under conditions where MLH1 was depleted, HER2 levels were shown to become elevated. Response to Fulvestrant (and subsequently, to other ER pathway inhibitors) was shown to occur in MLH1 depleted cells, in combination with a HER2 small molecule inhibitor. This was validated in the same cell line models, grown in 3D. A series of *in vivo* experiments are undertaken, including cell lines depleted for MLH1, which again show responses in the presence of both a HER2 targeted agent and an ER targeted drug. A number of PDX models with varying genetic and MLH1 levels were used to explore this treatment context.

This is a very interesting topic and the findings have the potential to be of substantial clinical relevance and as such, as a concept, I find the main conclusions to be important and novel. Most of the work is conducted to a high standard and although I understand that the authors conducted a number of different *in vivo* models, the critical comparison is whether the presence or absence of MLH1 impacts tumor growth and treatment response. This is inferred by using models with endogenously low or high MLH1 levels, but what is missing is the isogenically matched comparison. What is surprising is that the authors have these models, namely the shMLH1 and matched control cell line models, but this experiment is not conducted.

We are enthused that the reviewer finds the presented conclusions to be important and novel. We had originally not included the isogenic sh*Luc* control in the *in vivo* experiments because it is well established in the literature that parental MCF7 cells do not respond to lapatinib (PMID: 24554387, PMID: 17440164, PMID: 25249538). However, we now include the sh*Luc* control experiment in revised Fig 5 as described below.

- Why did the authors not repeat the experiment in Figure 4A (the MCF7 cell *in vivo* experiment) to compare the shMLH1 and the control line. Surely this is the most critical comparison, otherwise there is no *in vivo* data to show that MLH1 is important in the treatment response.
- What is concerning as well, is that the data presented from the *in vivo* experiment is a heavily modified version of the data and instead of simply plotting the tumor growth curves side by side, the authors present the data as 'slope of tumor growth' which is a non-standard way of presenting the data and is not intuitive or informative. The fact that the growth curves are not plotted according to standard approaches, leads me to conclude that the data wasn't very convincing. Given that this is the most important experiment (treatments to assess *in vivo* tumor growth in matched cells with or without MLH1), this piece of data is essential, but at the

moment, only one part of the data is presented and it is shown in a way that is opaque and uninformative, at best.

We have now conducted an MCF7 *shLuc* isogenic control xenograft experiment, presented in revised Fig 5B. As expected, in this experiment fulvestrant and estrogen deprivation alone completely inhibit tumor growth, and the addition of lapatinib to either arm has no additive or synergistic effect. These data are in keeping with the abundant literature in this regard, and we have described it in the revised text (pg 9, lines 7-17). We have also represented all tumor growth curves to show the raw data, rather than slope of tumor growth (revised Fig 5A-C).

- Why do the HER2 inhibitors only work in conditions where ER is inhibited. This seems like the key mechanistic finding, but no work (or even discussion) is provided. Why does ER need to be inhibited for HER2 targeted drugs to work under *MLH1* depleted conditions?

This is an extremely interesting question that we have been working on. We now present mechanistic data demonstrating that loss of ER upregulates HER2 in both *shLuc* and *shMLH1* cells. However, in *shLuc* cells this is accompanied by targeting of HER2 to lysosomes and subsequently to autophagosomes, whereas in *shMLH1* cells, HER2 remains at the membrane where it activates downstream signaling. We identified this mechanism through analysis of RNAseq (revised Fig 3A) and RPPA (revised Fig S4A) data from *shLuc* and *shMLH1*, MCF7 and T47D cells treated with fulvestrant, where we found significant differences in protein trafficking signatures between *shLuc* and *shMLH1* cells in both cell lines. We experimentally demonstrate this mechanism using

(1) time course co-immunofluorescence for LAMP1 (lysosomal marker) and HER2 in MCF7 *shLuc* and *shMLH1* cells after fulvestrant treatment (revised Fig 3B)

(2) chloroquine, an inhibitor of lysosomal trafficking in MCF7 *shLuc* cells (revised Fig 3C)

(3) live cell imaging for co-transfected LC3-RFP and HER-GFP plasmids in MCF7 *shLuc* and *shMLH1* cells treated with fulvestrant (revised Fig 3D).

We reproduce these results in T47D cells (revised Fig S4B-C).

These data now explain the need for ER inhibition, to increase baseline levels of HER2 in *shMLH1* cells. Description of these results are presented in revised text (pgs 5 (lines 17-20), 6 (lines 1-5) and pg 7 (lines 4-30)). Description of methods is presented in pg 14 (lines 6-16). Further discussion of these data is presenting in pgs 11 (lines 41-45) and 12 (lines 1-11)).

- Is *MLH1* regulated by estrogen stimulation or by the HER2 pathway? I am a little concerned that *MLH1* levels might correlate with outcome because it tracks with either ER and/or HER2 activity, rather than the other way around (i.e. that ER/HER2 activity and response to inhibitors is influenced by *MLH1* expression). The initial correlative data suggests that this is not the case, but as a control, it would be useful to assess whether *MLH1* is regulated by the ER and/or HER2 pathways.

ER inhibition using ER degrader, Fulvestrant, seems to moderately increase *MLH1* protein levels in our cell line models (presented below for the reviewer), which is consistent with the literature since ER promotes hypermethylation of the *MLH1* promoter (PMID: 29789325,

PMID: 32953793, PMID: 28323900). However, high ER levels (which would associate with low *MLH1* RNA) does not promote worse response to endocrine treatment or worse survival outcome (the opposite rather: PMID: 26455641), suggesting that *MLH1* is indeed the driver of the survival outcomes seen in patients, and not ER.

With HER2, we see no consistent effects on MLH1 either in patient data or in our model systems (transient transfection of MCF7 cells with siHER2 (revised Fig 4B) and treatment of T47D cells with lapatinib, a pan-HER inhibitor (revised Fig S3E)). These data, in addition to the observation that knocking down *MLH1*, either transiently (revised Fig S3D) or stably (revised Fig 2), induces HER2 upregulation that is reversible (revised Fig S3C), strongly suggests that MLH1 regulates HER2, rather than the other way around.

Reviewer #2 (Remarks to the Author):

Punturi et al have submitted an intriguing manuscript following on from previously published work showing the MLH1 loss drives endocrine treatment resistance. Here, they show that MutL loss activates HER2 in ER+/HER- breast cancer upon endocrine treatment and inhibiting HER2 can restore sensitivity in these tumours.

They present solid in vivo data confirming the potential therapeutic application of this data. However there are a number of issues that should be addressed prior to publication.

We thank the reviewer for their enthusiasm for the results presented here. In response to the reviewer, we have significantly revised the manuscript as described below.

1. There is little data investigating the mechanism behind HER2 activation upon MutL loss upon endocrine treatment. This needs further elucidation - how is HER expression regulated? Why is it ER dependent upon MutL loss?

This is an extremely interesting question that we have been working on. We now present mechanistic data demonstrating that loss of ER upregulates HER2 in both sh*Luc* and sh*MLH1* cells. However, in sh*Luc* cells this is accompanied by targeting of HER2 to lysosomes and subsequently to autophagosomes, whereas in sh*MLH1* cells, HER2 remains at the membrane where it activates downstream signaling. We identified this mechanism through analysis of RNAseq (revised Fig 3A) and RPPA (revised Fig S4A) data from sh*Luc* and sh*MLH1*, MCF7 and T47D cells treated with fulvestrant, where we found significant differences in protein trafficking signatures between sh*Luc* and sh*MLH1* cells in both cell lines. We experimentally demonstrate this mechanism using

(1) time course co-immunofluorescence for LAMP1 (lysosomal marker) and HER2 in MCF7 sh*Luc* and sh*MLH1* cells after fulvestrant treatment (revised Fig 3B)

(2) chloroquine, an inhibitor of lysosomal trafficking in MCF7 shLuc cells (revised Fig 3C)

(3) live cell imaging for co-transfected LC3-RFP and HER-GFP plasmids in MCF7 shLuc and shMLH1 cells treated with fulvestrant (revised Fig 3D).

We reproduce these results in T47D cells (revised Fig S4B-C).

These data now explain the need for ER inhibition, to increase baseline levels of HER2 in shMLH1 cells. Description of these results are presented in revised text (pgs 5 (lines 17-20), 6 (lines 1-5) and pg 7 (lines 4-30)). Description of methods is presented in pg 14 (lines 6-16). Further discussion of these data is presenting in pgs 11 (lines 41-45) and 12 (lines 1-11)).

2. There is no attempt to decipher whether this effect is mediated via MLH1 or PMS2, both components of the MutL complex and loss of MLH1 leads to reduced PMS2 levels - which would be restored upon MLH1 re-expression, so PMS2 alone as the mediator cannot be ruled out based on the current data.

This is an important question, and we now present data from shPMS2 MCF7 cells confirming that loss of PMS2 induces HER2 activation comparable to that seen in shMLH1 cells (revised Figs S6B-C), and concomitant sensitivity to the combination of HER inhibition and endocrine therapy (revised Fig S6D). Therefore, we conclude that both MLH1 and PMS2 loss can induce HER2 activation and sensitivity to HER2 inhibition, in keeping with patient dataset analyses in Figs 1&5. We describe the results in the revised text (pg 9, lines 23-28). This is also in keeping with our data from the WHIM20 PDX line which has intact MLH1 but is a PMS2 mutant.

3. The majority of the data are reliant on limited cell lines (only 2) and also no control cell lines.

We had originally not included the isogenic shLuc control in the *in vivo* experiments because it is well established in the literature that parental MCF7 and T47D cells do not respond to HER inhibition (PMID: 24554387, PMID: 17440164, PMID: 25249538). However, we now include an MCF7 shLuc isogenic control xenograft experiment, presented in revised Fig 5B. As expected, in this experiment fulvestrant and estrogen deprivation alone completely inhibit tumor growth, and the addition of lapatinib to either arm has no additive or synergistic effect. Since we also show these associations and phenotypes in multiple PDX lines (Fig 5) and in patient datasets (Figs 1&5), using orthogonal knockdown (revised Fig S3D) and rescue (revised Fig S3C), and in two related genes of the same complex (revised Fig S6B-D), we believe that the data is reliable.

4. The majority of the data is using only one shMLH1 to show the phenotype. It is widely known that gene silencing techniques in all forms have off target effects so it is imperative that additional approaches are used to validate the findings eg siRNA, CRISPR-Cas9, additional shRNA constructs

We have now included data demonstrating that transient transfection with orthogonal CRISPR sgRNA against MLH1 reproduces the phenotype for the shMLH1 cells (revised Fig S3D). In addition, data from an sh-resistant MLH1 rescue in shMLH1 lines (revised Fig S3C) and from the shPMS2 lines replicating the phenotype (revised Fig S6B-D) further increases our confidence that these results are not due to off target effects.

5. There is some inconsistency between Fig 2A and Fig3B in terms of pHER2 expression upon MLH1 loss. In Fig 2A there is visibly less pHER2 upon shMLH1 whilst no difference is observed in Fig 3B.

This inconsistency is because in Fig 2A, cells were treated with fulvestrant for 36 hours with no media change, while in original Fig 3B, media was changed 12 hours before lysates were collected, because of the requirements of the transient transfection protocol. The addition of fresh media (and therefore, growth factors), increases baseline HER2 in the sh*Luc* cells reproducibly in our hands. We provide here for the reviewer's reference, the complete Western blot, including the lysates collected at 24 hours (before the media change) where the difference in HER2 is clearly visible and comparable to Fig 2A.

6. Fig 3B needs an MLH1 blot to confirm silencing of the protein

We include Western blots for MLH1 in revised Fig 4B (original Fig 3B) to confirm silencing of the protein in our stably selected shMLH1 cells.

Reviewers' Comments:

Reviewer #1:

Remarks to the Author:

The authors have addressed my concerns and the inclusion of the new data and the full tumor growth curves satisfy my initial comments.

Reviewer #2:

Remarks to the Author:

The authors have adequately addressed all the points raised and provided significant additional data to back up their claims. I am happy to recommend publication